

**Dependence of the critical Richardson number on the temperature gradient in**
**the mesosphere**
Michael N. Vlasov and Michael C. Kelley
School of Electrical and Computer Engineering, Cornell University, Ithaca, NY 14853
Correspondence email: mv75@cornell.edu
**Abstract**
Maximum upper atmospheric turbulence results in the mesosphere from convective and/or
dynamic instabilities induced by gravity waves. For the first time, by comparing the vertical
accelerations induced by wind shear and the buoyancy force, it is shown that the critical
Richardson number $Ri_c$ can be estimated. Dynamic instability is developed for $Ri < Ri_c$. This
new approach, for the first time, makes it is possible to establish and estimate the temperature
gradient impact on dynamic instability development. Regarding our results, $Ri_c$ increases from
0.25 to 0.38 as the negative temperature vertical gradient increases from $\partial T/\partial z = 0$ to $\partial T/\partial z \leq$ -9
K/km. However, $Ri_c$ for the temperature, independent of altitude, is 0.25, coinciding exactly with
the $Ri_c$ commonly used and estimated in classical studies (Miles, 1961; Howard, 1961) and
subsequent papers without the temperature impact. The increase in the $Ri_c$ value strongly
influences cooling, inducing the cooling rate increase. Also, our results show that criterion $Ri_c <$
0.25 can only be used for the turbulent diffusion, which is characterized by eddies with sizes much
smaller than the scale height of the atmosphere. The $Ri_c$ value increases with the increasing size
of the eddies, but the term "eddy diffusion" cannot be applied to transport due to the large-scale
eddies (Vlasov and Kelley, 2015).



**Key Words:** 3334-Middle atmosphere dynamics, 3369-Thermospheric dynamics, 3379-
Turbulence

**1. Introduction**

In general, the Richardson number $Ri$ can be defined as the ratio of the destruction of turbulent

kinetic energy by buoyancy forces due to the production of turbulent energy by the wind shear
flow. This determination leads to the relation (see, for example, Peixoto and Oort (1992))
$$Ri = \omega_B^2/S^2 \, ,$$
(1)

where $\omega_B$ is the buoyancy frequency,
$$\omega_B^2 = \frac{g}{T}\left(\frac{\partial T}{\partial z} + g/C_p\right) ,$$
(2)

and $T$ is the temperature, $g$ is the acceleration of gravity, $C_p$ is the heat capacity at constant pressure,
and
$$S = \frac{\partial V}{\partial z}$$
(3)

is the vertical shear of the horizontal wind with the velocity $V(z)$ height profile. It is generally
accepted that a dynamic instability develops when the Richardson number is less than ¼, i.e., the
parcel's vertical motion induced by wind shear dominates the motion induced by the buoyancy
force. The former creates and the latter destroys these perturbations. Most authors use the critical
Richardson number $Ri_c < $ ¼ without references. Some authors refer to Miles (1961) and Howard
(1961). They consider the stable-stratified, horizontal shear flows of an ideal fluid. A set of studies
takes into account the time-dependent shear flow and the results of laboratory experiments
(Peixoto and Oort, 1992; Galperin et al., 2007). However, we could not find papers on the critical
Richardson number that take the mesospheric conditions into account. Miles and other authors
(Abarbanel et al., 1984; Ligniéres et al., 1999; Galperin et al., 2007) did not consider the



temperature's influence on the $Ri_c$ value. However, the eddy turbulence peak is observed in the
mesosphere or the lower thermosphere where the large negative and positive gradients of the
temperature occur. We could find just one paper [Hysell et al., 2012] on the estimate of the $Ri_c$
value in the lower thermosphere. Using the data on observations of the sporadic $E$ layer, Hysell et
al. (2012) inferred the parameters of wind shear corresponding to the irregularities observed in the
layer and estimated the $Ri_c$ value of 0.75. However, the authors used the wrong formula for the
background density, resulting in densities much larger than the observed atmospheric density
corresponding to the hydrostatic equilibrium. It is shown in Appendix 3 how $0.7 < Ri_c < 0.8$ can
be found due to the background density used by Hysell et al. (2012).

The principal measure of stability regarding the buoyancy effects of the density gradient

overriding its inertial effects is the Richardson number given by formula (1) in Miles (1961), which
can be written as
$$Ri = -g\frac{\partial \rho}{\partial z} \Big/ \left\{\rho \left[\frac{\partial V}{\partial z}\right]^2\right\}, \tag{4}$$

where $\rho$ is the density and $V$ is the horizontal wind velocity. This formula can be rewritten as
$$\left(\frac{\partial V}{\partial z}\right)^2 = -\frac{g}{Ri}\frac{1}{\rho}\frac{\partial \rho}{\partial z}. \tag{5}$$

This initial formula will be used here to estimate the accelerations induced by wind shear and the
buoyancy forces under mesospheric conditions.

The goal of this paper is to estimate the critical Richardson number, $Ri_c$, corresponding to the

equilibrium between the buoyancy force and the force induced by wind shear in the mesosphere.

Dynamic instability is developed for $Ri < Ri_c$. Our approach considers the acceleration

corresponding to both forces, taking into account the mesospheric temperature height distributions.





**2. Acceleration Induced by Wind Shear**
We start from formula (5) corresponding to the initial equation used by Miles (1961) (here,
formula (4)). Miles considers an uncompressible fluid but the adiabatic expansion/compression
should be taken into account in the upper atmosphere. Differentiating the adiabatic relation
$pT^{-\gamma/(\gamma-1)} = const$ corresponding to Poisson's equation where $p = \rho\kappa T/m$ and $p$ is the pressure;
$m$ is the mean molecular mass; $\gamma = C_p/C_v$; $C_p$ and $C_v$ are the heat capacities at constant pressure
and volume; $\gamma/(\gamma-1) = 1 + N/2$; $N = 5$ is the number of degrees of freedom for diatomic gas;
and $\kappa$ is the Boltzmann's constant, it is possible to get the adiabatic expansion equation
$$\frac{1}{\rho}\frac{\partial\rho}{\partial z} = \frac{N}{2}\frac{1}{T}\frac{\partial T}{\partial z} \qquad (6)$$
(see the derivation of this formula in Appendix1), and according to formula (5):
$$\left(\frac{\partial V}{\partial z}\right)^2 = -\frac{g}{Ri}\frac{N}{2T}\frac{\partial T}{\partial z} \ . \qquad (7)$$
Taking into account $Ri(\partial V/\partial z)^2 = \omega_B^2 = (g/T)(\partial T/\partial z + g/C_p)$ and using formula (6), the
temperature gradient in the parcel with upward motion and adiabatic expansion can be given by
the equation
$$\frac{\partial T}{\partial z} = -\frac{g}{(1+N/2)C_p} \qquad (8)$$
and
$$T = T_0 - \frac{g}{\left(1+\frac{N}{2}\right)C_p}(z - z_0) \ . \qquad (9)$$

By substituting formulas (8) and (9) in formula (7) multiplied by $(z - z_0)$, it is possible to
obtain the formula
$$a_{ws} = \frac{g^2 N(z-z_0)}{2Ri[T_0 C_p(1+N/2) - g(z-z_0)]} \qquad (10)$$



where
$$a_{ws} = \left(\frac{\partial V}{\partial z}\right)^2 (z - z_0) \qquad (11)$$

is the acceleration in wind shear. As can be seen from Fig. 1, this acceleration increases with the
increase of the vertical size of the wind shear layer. Note that this size cannot exceed 1–2 km
according to the experimental data (Larsen, 2002). The $a_{ws}$ dependence on the altitude is linear
because $g(z - z_0) \ll T_0 C_p (1 + N/2)$ for $-z_0 < 2$ km.

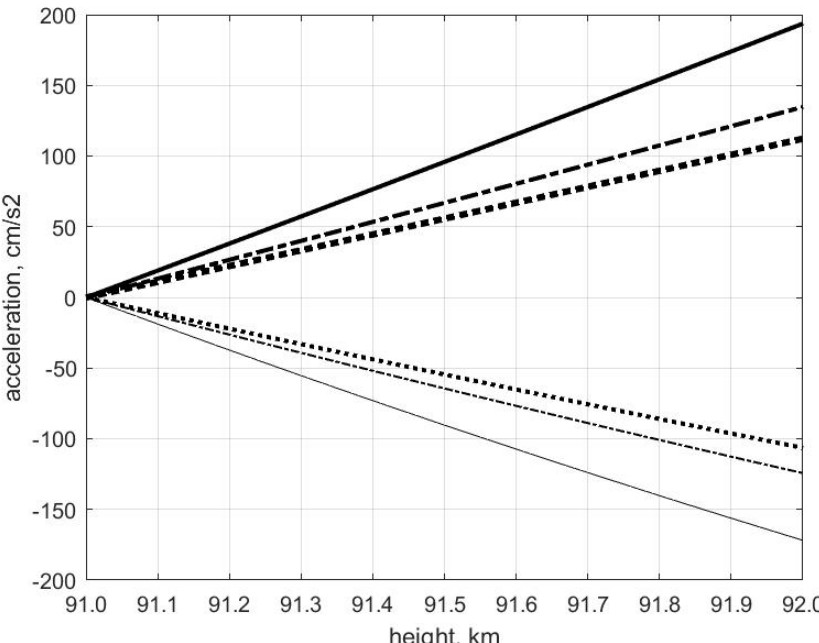


**Figure 1.** The height profiles of the wind shear $a_{ws} > 0$ and buoyant $a_B < 0$ accelerations calculated
by formulas (11) and (15), respectively, with $T_0 = 140$ K and $Ri_c = 0.25$ (solid curves), with $T_0 = $
K and $Ri_c = R/C_p = 0.286$ (dashed-dotted curves), and with $T_0 = 200$ K and $Ri_c = 0.286$
(dotted curves).





**3. Acceleration Induced by the Buoyancy Force**
The buoyancy force is $F_B = g(\rho_A - \rho_D)$ where $\rho_A$ and $\rho_D$ are the background atmospheric
density and the disturbed density, respectively. The acceleration is given by
$$a_B = g[(\rho_A - \rho_D)\rho_D] \,. \tag{12}$$

The atmospheric density distribution can be given by
$$\rho_A = \rho_{A0} exp[-(z - z_0)/H_A] \tag{13a}$$

for $dT_A/dz = 0$ in the mesopause and the formula
$$\rho_A = \rho_{A0}\{[T_{A0} - G(z - z_0)]/T_{A0}\}^{(mg/\kappa G - 1)} \tag{13b}$$

for $dT_A/dz = G < 0$ below the mesopause, and $H_A = \kappa T_{A0}/mg$ is the scale height of the
atmospheric gas. By integrating equation (6) with the temperature and temperature gradient given
by formulas (8) and (9), it is possible to get the disturbed density distribution ($T_0 = T_{A0}$),
$$\rho_D = \rho_{A0}\left[\frac{T_0 - \frac{G(z-z_0)}{C_p(1+N/2)}}{T_0}\right]^{N/2}, \tag{14}$$

and the acceleration corresponding to the buoyancy force can be written as
$$a_B = g\left[\left(\frac{\rho_A}{\rho_D}\right) - 1\right] = g\frac{\rho_{A0}e^{-\frac{(z-z_0)}{H_A}}}{\rho_{A0}\left[\frac{T_0 - \frac{g(z-z_0)}{C_p(1+\frac{N}{2})}}{T_0}\right]^{\frac{N}{2}}} - g \tag{15}$$

for $dT_A/dz = 0$. As seen from Fig. 1, there is very good agreement between the $a_{ws}$ and $a_B$
absolute values for $Ri_c = 0.25$, and $T_0 = 140$ K and $T_0 = 200$ K for the vertical size of a stable wind
shear layer that is less than 400 m. The $a_{ws}$ value becomes larger than the $a_B$ value for $z - z_0 >$
400 m, which means that the $Ri_c$ value should be increased. The turbulence develops if $\alpha_{ws}$ is
larger than the $\alpha_B$ that corresponds to $Ri < Ri_c$. We emphasize that the perturbation scale sizes



induced by wind shear do not exceed 1-2 km, according to the observations (see Lübken (1997)).
Note that formula (13b) should be used instead of formula (13a) in the nominator of formula (15)
for atmospheric temperature distribution with $\frac{dT_A}{dz} < 0$. As can be seen from Fig. 2, the $a_B$ values
significantly decrease in this case, since the atmospheric density given by formula (13b) is larger
and the density gradient is less than the density and gradient corresponding to formula (13a). The
small buoyancy force corresponds to the small density gradient. This dependence explains the $a_B$
reduction with the $T_A$ decrease.

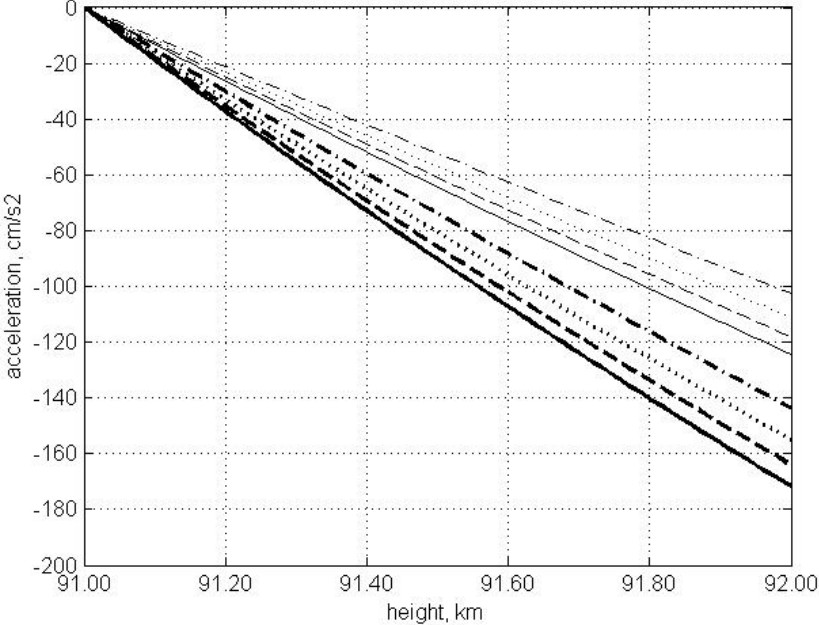


**Figure 2.** The height profiles of the acceleration of the buoyancy force calculated by formula (15)
with the nominator $\rho_{A0}\{[T_{A0} - G(z - z_0)]/T_{A0}\}^{(mg/\kappa G-1)}$ for $T_0 = T_{A0} = 140$ K and 200 K (thick
and thin curves, respectively) and $G = 1, 2.8,$ and 5 K/km (dashed, dotted and dashed-dotted curves,
respectively), and calculated by formula (15) (solid curves).






## 4. Estimating the Richardson Number


Using formulas (11) and (15) in the equation $a_{ws} + a_B = 0$, the formula for $Ri_c$ can be inferred:
$$Ri_c = \frac{\left[1-\frac{g(z-z_0)}{T_0 C_p(1+N/2)}\right]^{N/2}\frac{gN(z-z_0)}{2C_p(1+N/2)\left[T_0-\frac{g(z-z_0)}{C_p(1+N/2)}\right]}}{\left[1-\frac{g(z-z_0)}{T_0 C_p(1+N/2)}\right]^{N/2}-exp\left[-\frac{(z-z_0)}{H_A}\right]} . \tag{16}$$

The $Ri_c$ values calculated by formula (16) and this formula with $\{[T_0 - G(z - z_0)]/T_0\}^{(mg/\kappa G-1)}$
(see formula (13b) instead of the exponential term) are shown in Figs. 3a and 3b. The $Ri_c$ values
increase with increasing altitude, corresponding to the vertical expansion of the region of the stable
wind shear. However, according to the experimental data (Larsen, 2002; Kelley et al., 2003;
Bishop et al., 2004), the wind shears are very unstable. As mentioned above, the size scales of the
density perturbations do not exceed 1 – 2 km, according to the observations. A more accurate
consideration of eddy turbulence (Vlasov and Kelley, 2015) concludes that the scale size of density
perturbations *l* should be much less than the scale height of atmospheric gas, $l << H_A$ and $l << 4$
km for $T_A = T_0 = 140$ K and $l << 5.7$ km for $T_A = T_0 = 200$ K. However, this restriction can only
apply to turbulence corresponding to the eddy diffusion approximation (Vlasov and Kelley, 2015).
As seen from Fig. 3a, the $Ri_c$ value of 0.25 corresponds to perturbations with scales less than 10
m, and the $Ri_c$ values reach 0.256 and 0.263 for $l = 200$ m and 400 m and for $T_{A0} = 140$ K and
0.254 and 0.257 for $T_0 = 200$ K, respectively. The $Ri_c$ value of 0.25 corresponds to the mean
value $l = 27.3$ m obtained by Lübkin (1997), using the measured spectrum of the density
fluctuation. Vlasov and Kelley (2015) reconsidered the results of Kelley et al. (2003) and found
that the spectrum scale fluctuations inferred from the meteor train turbulence observations can be
approximated by Heisenberg's formula with $l = 119$ m, and eddies with very large scales may



occur in the narrow layer of localized turbulence. As can be seen from Fig. 3b, the $Ri_c$ values
increase with the increase in the negative gradient of the temperature and can reach almost 0.36.

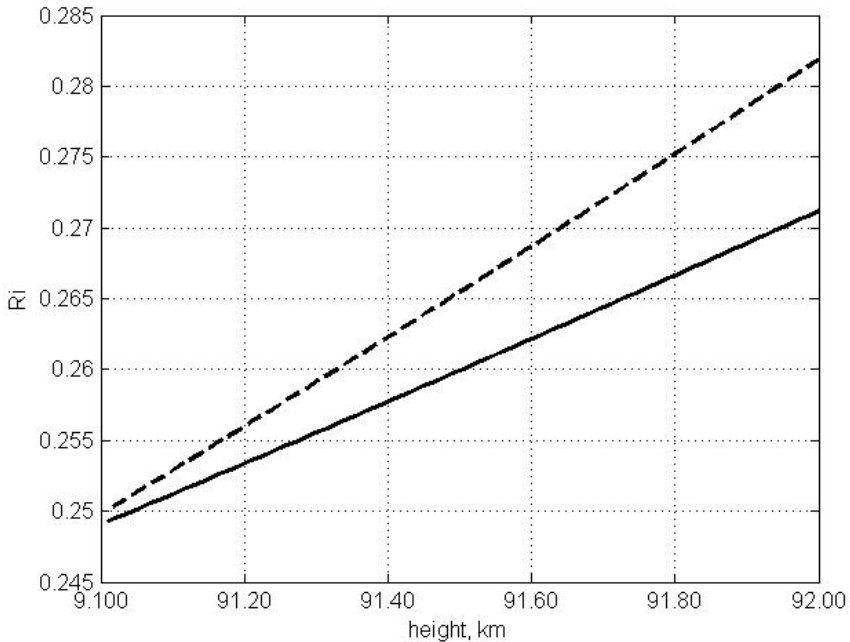


**Figure 3a.** The height profiles of the critical Richardson number calculated by formula (16) with
$T_0$ = 140 K and 200 K (dashed and solid lines, respectively).





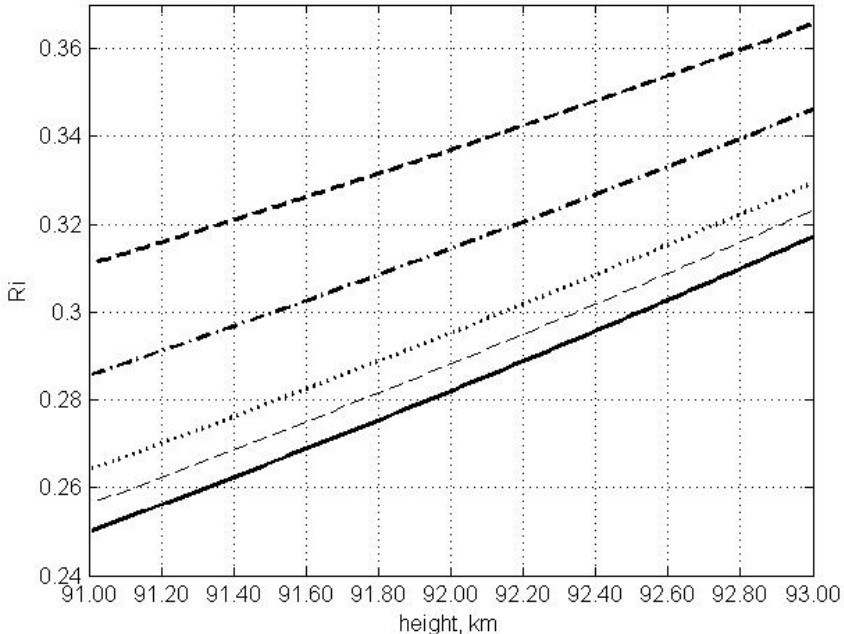


**Figure 3b.** The height profiles of the critical Richardson number calculated by formula (16) with

$\{[T_0 - G(z - z_0)]/T_0\}^{(mg/\kappa G - 1)}$ instead of the exponential term for the $T_0 = 140$ K with $dT/dz =$

$G < 0$ with $|G| = 0.2, 1, 3$, and 5 K/km (dashed thin, dotted, dashed-dotted and dashed thick curves,

respectively) and calculated by formula (16) (solid thick curve).

Thus, turbulence can develop with $Ri_c > 0.25$ for wind shears with a vertical size of 1–2 km,

but this turbulence may not correspond to eddy diffusion. The scales of the density fluctuations

are very small (for example, see Lübken (1997)) that correspond to $z \rightarrow z_0$. However, the $Ri_c$ value

estimation for $z \rightarrow z_0$ is problematic because, in this case, the numerator and denominator in

formula (16) try to attain zero. This uncertainty can be solved using L'Hospital's rule, leading to

the formula (see Appendix 2)



$$Ri_c = \frac{0.5gN}{g(1+N/2)^2 - 0.5gN - GC_p(1+N/2)}$$
(17)

for the $Ri_c$ limit value for $z \to z_0$. This formula corresponds to the limit value formula (16) with
the term $\{[T_0 - G(z - z_0)]/T_0\}^{(mg/\kappa G - 1)}$ instead of the term $exp[-(z - z_0)/H_A]$. The $Ri_c$
dependence on the negative temperature gradient, given by formula (17), is shown in Fig. 4. The
$G$ increase improves the conditions for the dynamic instability development. Note that the $Ri_c$
value for $G = 0$ coincides with the results of Miles (1961) and the commonly used value of $Ri_c$.

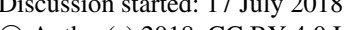


**Figure 4.** The dependence of the Richardson number $Ri_c$ on the temperature negative gradient
calculated by formula (17).

**5. The Influence of $Ri_c$ Dependence on $G$ on Cooling in the Mesosphere**



The eddy turbulence heating/cooling rate can be given by the equation (Vlasov and Kelley,

2010)

$$Q_{ed} = \frac{\partial}{\partial z}\left[K_{eh}C_p\rho\left(\frac{\partial T}{\partial z} + \frac{g}{C_p}\right)\right] + K_{eh}\rho\frac{g}{Tb}\left(\frac{\partial T}{\partial z} + \frac{g}{C_p}\right),$$    (18)

where $K_{eh}$ is the coefficient of the eddy heat transport, $\rho$ is the undisturbed gas density, and $b$ is a
dimensionless constant given by the relation obtained using the results of Gordiets et al. (1982),

$$b = Ri_c/(P - Ri_c)$$    (19)

where $P$ is the turbulent Prandtl number. According to equation (18), the $Q_{ed}$ value is given in units
$erg \times cm^{-3} \times s^{-1}$. The $K_{eh}$ value is given by

$$K_{eh} = b\varepsilon/\omega_B^2,$$    (20)

where $\varepsilon$ is the energy dissipation rate, and $b$ can be given by formula (19). The vertical distribution
of the $\varepsilon$ value in the turbulent layer can be approximated by the Gaussian function

$$\varepsilon = \varepsilon_m exp[-(z - z_m)^2/h^2] ,$$    (21)

where $h$ is half of the layer thickness and $\varepsilon_m$ is the $\varepsilon$ value at the altitude of the layer peak $z_m$.
Using this approximation, dividing equation (18) by $\rho C_p$ and substituting formula (20) with $b =$
$Ri/(P - Ri)$ and $T = T_0 + G(z - z_0)$, equation (18) can be written in units $K/s$ as

$$Q_{ed} = \varepsilon_m exp\left[-\frac{(z-z_m)^2}{h^2}\right]\left\{\frac{[T_0+G(z-z_0)]}{g\left(\frac{P}{Ri_c}-1\right)}\left[-\frac{2(z-z_m)}{h^2} - \frac{\frac{mg}{\kappa}}{T_0+G(z-z_0)}\right] + \frac{1}{C_p}\right\}.$$    (22)

Using the $Ri_c$ dependence on the temperature gradient given by formula (17), the impact of the
Richardson number on the cooling rates can be estimated. According to the results in Fig. 5, the
cooling rates increase by a factor of 2.2 for $0.25 < Ri_c < 0.38$ corresponding to $0 \leq G \leq$ -9 K/km,
but the $G$ value influence on the cooling for $Ri_c$ = const = 0.25 is very small (curves near the thick
solid curve). Note that the turbulence induced by the large wind shear may not correspond to the




eddy diffusion heat transport. The values of $\varepsilon_m$, $z_m$, and $h$ correspond to the experimental data
(Lübken, 1997).

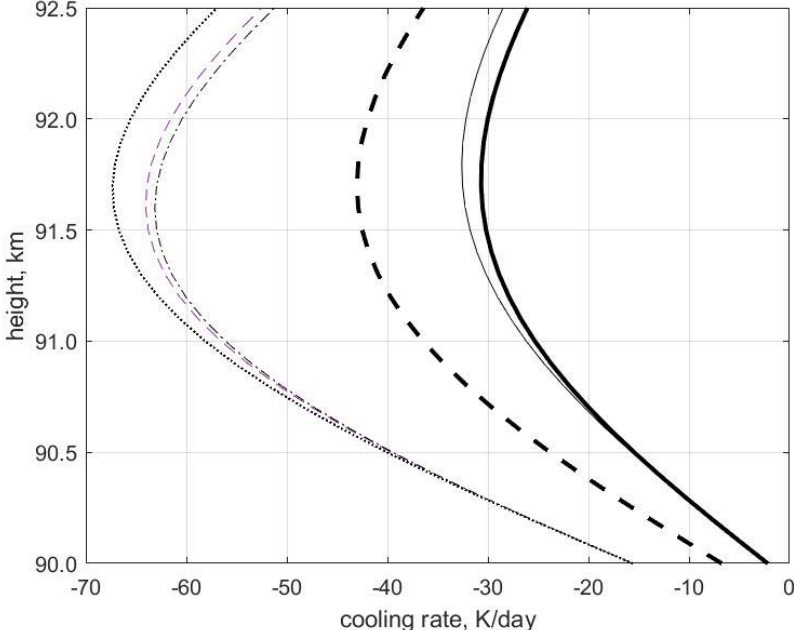


**Figure 5.** The cooling rates calculated by equation (22) with $G = 0$ K/km – $Ri = 0.25$, $G = -3$ K/km
– $Ri = 0.286$, $G = -5$ K/km – $Ri = 0.31$, G = -7 K/km – $Ri = 0.34$, $G = -8$ K/km – $Ri = 0.36$, G = -9
K/km – $Ri = 0.38$ (thick solid, dashed and dashed-dotted curves and thin dotted, solid curves and
thick dotted curve, respectively) and the $Q_{ed}$ values calculated with $Ri = 0.25$ and the $G$ values
from -3 K/km to -9 K/km are shown by curves near the thick solid curve.

**6.   Conclusions**

For the first time, by comparing the accelerations in wind shear and the buoyancy force, it is

shown that the critical Richardson number, corresponding to the equilibrium of these forces, can



221 be estimated and the dynamic instability developed for $Ri < Ri_c$. This new approach is very

222 different from the approach used in classical studies (Miles, 1961) and subsequent papers. Note

223 that Miles and the other authors did not consider the temperature's influence on dynamic instability

224 development. However, the mesosphere is characterized by the negative temperature gradient, and

225 the turbulence peak is observed in this region. For the first time, it has been estimated and

226 established that the $Ri_c$ value depends on the temperature gradient. The $Ri_c$ value increases with

227 the negative mesospheric temperature gradient increase. It should be emphasized that our

228 estimated $Ri_c$ value is exactly the same as the $Ri_c$ value of 0.25 estimated by Miles (1961) and

229 other authors and does not depend on the temperature for $dT/dz = 0$.

230 The Richardson number dependence on the temperature gradient influences the cooling rates

231 induced by eddy turbulence. These rates significantly increase with an increasing $Ri_c$, but the

232 influence of the negative temperature gradient on the cooling for $Ri_c = const = 0.25$ is very

233 small.

234 Also, our results show that criterion $Ri_c = 0.25$ can be used for turbulent diffusion that is

235 characterized by eddies with a size that is much less than the scale height of the atmosphere. The

236 $Ri_c$ value increases with the increase in the vertical size of the wind shear (see Fig. 3a), but there

237 is a problem with applying the term "eddy diffusion" to momentum and heat transport because of

238 the large-scale eddies in this case (Vlasov and Kelley, 2015).

239 In general, our results show that the criterion $Ri_c = 0.25$ can only be applied to turbulence with

240 small scales corresponding to the eddy diffusion. This diffusion provides the mixing of neutral

241 constituents and their diffusive separation as a result of the competition between eddy and

242 molecular diffusion. In this case, the criterion $Ri_c = 0.25$ is necessary and sufficient, but not for

243 the more complicated shears mentioned above and observed in the lower thermosphere.



**Appendix 1**

Derivation of formula (6) in the paper. We start by using the adiabatic equation $pT^{-\gamma/(\gamma-1)} = const$:

$$\frac{\partial}{\partial z}\left[pT^{-\gamma/(\gamma-1)}\right] = 0 \tag{A1}$$

$$p = \rho R T \tag{A2}$$

$$\gamma = Cp/Cv = 1 + 2/N \tag{A3}$$

$$\gamma/(\gamma-1) = 1 + N/2 \tag{A4}$$

$$\frac{\partial}{\partial z}\left[R\rho T \times T^{-1-N/2}\right] = R\left[\frac{\partial\rho}{\partial z}T^{-N/2} - \rho\frac{N}{2}T^{-1-N/2}\frac{\partial T}{\partial z}\right] = 0 \; . \tag{A5}$$

Dividing this equation by $\rho$ and multiplying by $T^{-N/2}$, it is possible to get the adiabatic expansion equation

$$\frac{1}{\rho}\frac{\partial\rho}{\partial z} = \frac{N}{2}\frac{1}{T}\frac{\partial T}{\partial z} \; . \tag{A6}$$

**Appendix 2**

Derivation of formula (17) for $\partial T/\partial z = G = 0$:

$$Ri_c = \frac{\left[1-\frac{g(z-z_0)}{B}\right]^{N/2}}{\left[1-\frac{g(z-z_0)}{B}\right]^{N/2}-exp\left[-\frac{(z-z_0)}{H_A}\right]}\frac{0.5gN(z-z_0)}{B-g(z-z_0)} = \frac{F(z)}{\varphi(z)} \tag{A1}$$

where $B = T_0 C_p(1 + N/2)$ and

$$\frac{\partial F}{\partial z} = -\frac{Ng}{2B}\left[1-\frac{g(z-z_0)}{B}\right]^{N/2-1}\frac{0.5gN(z-z_0)}{B-g(z-z_0)} + \left[1-\frac{g(z-z_0)}{B}\right]^{N/2}\frac{0.5gN[B-g(z-z\_0)]+0.5gN(z-z_0)g}{[B-g(z-z_0)]^2} \; .$$

$$\tag{A2}$$

For $z = z_0$,

$$\frac{\partial F}{\partial z} = \frac{0.5gNB}{B^2} = \frac{0.5gN}{B} \tag{A3}$$





$$\frac{\partial \emptyset}{\partial z} = -\frac{N_g}{2B}\left[1 - \frac{g(z-z_0)}{B}\right]^{N/2-1} + \frac{1}{H_A} exp\left[-\frac{(z-z_0)}{H_A}\right].$$
(A4)

For $z = z_0$,
$$\frac{\partial \emptyset}{\partial z} = -\frac{Ng}{2B} + \frac{1}{H_A}.$$
(A5)

Finally, we have a very simple formula:
$$Ri = \frac{0.5gN}{B\frac{mg}{\kappa T_0}-0.5gN} = \frac{0.5N}{\left(1+\frac{N}{2}\right)^2-0.5N} = 0.256 \text{ for } N = 5, G = 0$$
(A6)

and for $G< 0$,
$$\frac{\partial \emptyset}{\partial z} = -\frac{0.5N_g}{B} - \frac{\partial}{\partial z}\left\{\frac{[T_0-G(z-z_0)]}{T_0}\right\}^{\frac{mg}{\kappa G}-1} = -\frac{0.5N_g}{B} - \left(\frac{mg}{\kappa G}-1\right)\left(\frac{-G}{T_0}\right) \text{ for } z = z_0$$
(A7)

$$\frac{\left(\frac{\partial F}{\partial z}\right)}{\left(\frac{\partial \emptyset}{\partial z}\right)} = \frac{0.5gN}{B\left[-\frac{0.5gN}{B}+\frac{mg}{\kappa T_0}-\frac{G}{T_0}\right]} = -\frac{0.5gN}{-0.5gN+g\left(1+\frac{N}{2}\right)^2-\frac{GB}{T_0}} = \frac{0.5gN}{\left(1+\frac{N}{2}\right)^2 g-0.5N_g-GC_p(1+N/2)}.$$
(A8)


**Appendix 3**
The equation used by Hysell et al. (2009, 2012) is
$$N^2 = -\frac{g}{\rho_0}\frac{\partial \rho_0}{\partial z} = \frac{g}{T}\left(\frac{\partial T}{\partial z}+\frac{g}{C_p}\right).$$
(A1)

Here, $N^2$ is the buoyancy frequency square and $\rho_0$ is the background density. This equation is
incorrect because first, the buoyancy frequency for incompressible fluid is not equal to the
frequency for compressible fluid, and second, the background density given by the equation
$$\frac{1}{\rho_0}\frac{\partial \rho_0}{\partial z} = -\frac{1}{T}\left(\frac{\partial T}{\partial z}+\frac{g}{C_p}\right)$$
(A2)

is much larger than the density given by the equation
$$\frac{1}{\rho_A}\frac{\partial \rho_A}{\partial z} = -\frac{1}{T}\left(\frac{\partial T}{\partial z}+\frac{g}{R}\right)$$
(A3)




for hydrostatic equilibrium corresponding to real atmospheric conditions. For example, the scale
height of the density is $H = \kappa T(1 + N/2)/mg$ corresponding to equation (A2) where $\partial T/\partial z = 0$
is larger by a factor of 3.5 than the scale height of the background atmospheric density $H =$
$\kappa T/mg$ corresponding to equation (A3). The atmospheric density inferred from equation (A2)
with $\partial T/\partial z = G$ is given by the formula
$$\rho_A = \rho_{A0}\{[T_{A0} + G(z - z_0)]/T_{A0}\}^{(-mg/\kappa G(1+0.5N)-1)}.$$    (A4)
This formula is similar to formula (13b) but with $G > 0$ and $-mg/\kappa G(1 + 0.5N)$ instead of
$-mg/\kappa G$. The density given by formula (A4) is much larger than the density given by formula
(13b) for $G > 0$. Substituting formula (A4) instead of the exponential term in equation (16) and
using L'Hospital's rule, it is possible to get the equation
$$Ri_c = \frac{0.5gN}{g(1+0.5N)-0.5gN+GC_p(1+0.5N)} = \frac{0.5gN}{g+GC_p(1+0.5N)}$$    (A5)
instead of equation (17).
According to Fig. 2 in Hysell et al. (2012), a sporadic $E$ layer with significant irregularities was
observed by Arecibo INR at a height of around 110 km at 19:30 – 20:30 LT on July 2, 2010 in the
lower thermosphere. The authors used the data on this layer to infer the parameters of the wind
shear and then, using a numerical model, they estimated the $Ri_c$ value of 0.75 for the dynamic
instability corresponding to the observed irregularities in this region. According to the data shown
in Fig. 2 (Hysell et al., 2012), the temperature gradient in the instability at around 110 km is $G =$
6-8 K/km and the $Ri_c$ value can be found to be $0.8 - 0.65$, respectively, according to equation
(A5). It follows that the large $Ri_c$ value of 0.75 estimated by the numerical model of Hysell et al.
(2012) can only result from the large density used instead of the correct background density. In
this case, the $Ri_c$ value does not depend on the specific features of wind shear inferred by the
authors and used in the numerical model. According to equation (17) with $G > 0$ and the



background density given by formula (13b) with $G > 0$, the $Ri_c$ value decreases from 0.25 to 0.2
with $G$ increasing from 0 to 8 K/km.

**Competing Interests**
The authors declare that they have no conflict of interest.

**Acknowledgments**
Work at Cornell University was funded by the School of Electrical and Computer Engineering
and the Cornell Podell Emeriti Awards for Research and Scholarship (PEARS) Program through
CAPE. This paper is entirely theoretical and no data have been used.

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
