# Peer review of "Dependence of the critical Richardson number on the temperature gradient in"

_Annales Geophysicae, 2018_

## Referee Comment (RC1) · Anonymous Referee #1 · 21 Sep 2018

**Comments on the paper "Dependence of the critical Richardson number on the temperature gradient in the mesosphere" by Michael N. Vlasov and Michael C. Kelley submitted for publication in ANGEO ($2016JD025811$)**

In the paper authors described their simulations of dependence of the critical Richardson number on temperature gradient in the mesosphere.

I mainly concerned with the question: What is importance (or application) of this work?

From theoretical studies and laboratory experiments, the critical Richardson number ($Ri_c$) has been determined to be approximately, **but not precisely** equal to 0.25 [e.g., *Grachev et al.*, 2012, and references therein]. For example, *Haack et al.* [2014] observed large variation of energy dissipation rate for wide range of $Ri$-values, i.e. also for $Ri > 1$. In regions where $Ri$ was $> 1$ and far beyond, turbulent layers have been detected. Thus, the increase of $Ri_c$ from 0.25 to 0.38 found in this work is by no means crucial, especially if we take into account very limited altitude range (within mesosphere-lower thermosphere region) used by authors for simulations.

List of references contains only 14 items (with 2 from the same authors). That is, a proper overview of previous results regarding $Ri_c$ is not present. Quick look in 50 years old overview articles [e.g., *Reiter and Lester*, 1967; *Obukhov*, 1971] reveals that different values of $Ri_c$ with different approaches were obtained, which is not mentioned in this manuscript. In the manuscript authors refer to papers of *Miles* and *Howard* from 1961 (i.e., 57 years back). Does it mean, that in the last half century nobody considered this problem? Also some reference to the statement "*However, the eddy turbulence peak is observed in the mesosphere or the lower thermosphere where the large negative and positive gradients of the temperature occur.*" (lines 47-49) is needed.

Authors concluded, that *$Ri_c$ value depends on the temperature gradient. The $Ri_c$ value increases with the negative mesospheric temperature gradient increase.* (for example, lines 226-227). These statements were supported by different figures that only show an altitude range above 90 km where temperature gradient is already positive. This is confusing. Is it possible to use altitude range below mesopause for simultions?

**References**

Grachev, A., E. L Andreas, C. Fairall, P. Guest, and O. Persson, The critical richardson number and limits of applicability of local similarity theory in the stable boundary layer, *147*, 2012.

Haack, A., M. Gerding, and F.-J. Lübken, Characteristics of stratospheric turbulent layers measured by LITOS and their relation to the Richardson number, *J. Geophys. Res.*, pp. 10,605–10,618, doi:10.1002/2013JD021008, 2014.

Obukhov, A. M., Turbulence in an atmosphere with a non-uniform temperature, *Boundary-Layer Meteorology*, *2*, 7–29, doi:10.1007/BF00718085, 1971.

Reiter, E. R., and P. F. Lester, The Dependence of Richardson Number on Scale Length, *Colorado State University, Atmospheric Science*, *111*, 39, 1967.

---

## Author Comment (AC1) · 1 Nov 2018

Response to the Comments of Reviewer 1 Regarding the Paper:

"Dependence of the critical Richardson number on the temperature gradient in the mesosphere"
by Michael N. Vlasov and Michael C. Kelley

Submitted for Publication in *Annales Geophysicae* (ANGEO 2016JD025811)

*In the paper authors described their simulations of dependence of the critical Richardson number on temperature gradient in the mesosphere. I mainly concerned with the question: What is importance (or application) of this work?*

The critical Richardson number $Ri_c$ is the criterion for the development of turbulence as a result of the dynamic instability induced by gravity waves. The reviewer emphasizes an uncertainty in the estimate of this parameter (see the reviewer's comment below). We believe that, in this case, the estimate of the $Ri_c$ value using the new independent approach is very important. Also, for the first time, this new approach makes it is possible to establish and estimate the dependence of the $Ri_c$ value on the temperature negative gradient. Note that according to our results, the $Ri_c$ value does not depend on the temperature for $\partial T/\partial z = 0$. These new results also broaden our understanding of the role of turbulence in the heat balance in the mesosphere. It is shown by us that the $Ri_c$ increase with the increasing temperature negative gradient induces a rise in the cooling rate in the mesosphere. Our results can also be used to estimate the diffusion coefficient of the eddy turbulence from the experimental data on the energy dissipation rate of the gravity waves. The total density of the thermosphere depends on this coefficient (see our next comment).

It should be emphasized that the temperature negative gradient is the main feature of the mesosphere. This region is very important because of the transition from the *homosphere* to the *heterosphere*. The former corresponds to the mixing process due to turbulence and the latter corresponds to the diffusive separation of the atmospheric constituents. The maximum turbulence takes place in the mesosphere due to the peak of the gravity wave dissipation.

*From theoretical studies and laboratory experiments, the critical Richardson number (Ric) has been determined to be approximately, but not precisely equal to 0:25 [e.g., Grachev et al., 2012, and references therein]. For example, Haack et al. [2014] observed large variation of energy dissipation rate for wide range of Ri-values, i.e. also for Ri > 1. In regions where Ri was > 1 and far beyond, turbulent layers have been detected. Thus, the increase of Ric from 0:25 to 0:38 found in this work is by no means crucial, especially if we take into account very limited altitude range (within mesosphere-lower thermosphere region) used by authors for simulations.*

It is well known that there is a problem with determining the $Ri_c$ value. For example, Weinstock [1978] assumed that the turbulence produced in regions of dynamic instability ($Ri_c < 0.25$) could be transported by turbulent flux into regions of larger $Ri$ and the $Ri_c$ mean value might be 0.44.

Galperin et al. [2007] considered a spectral theory of turbulence that accounts for strong anisotropy and waves. They lead to the conclusion that "the effects of the nonstationarity, internal waves, and strong anisotropization preclude the laminarization of turbulence and thus make $Ri_c$ devoid of its conventional meaning". However, these authors do not take into account the different types of turbulence and the conditions for their development. For example, they do not distinguish between uniform and localized turbulence. The latter is additionally characterized by the Prandtl number. Also, there are different types of wind shear and their interaction with buoyancy forcing, and the latter depends on the temperature gradient. The observations of unstable layers during the Turbulent Oxygen Mixing Experiment (TOMEX) showed the very large values of the energy dissipation rates (0.9 W/kg) and the eddy diffusion coefficients inferred from these data were found to be 1900 m$^2$/s by Bishop et al. [2004]. However, Vlasov and Kelley [2015] showed that these large values cannot be considered as the eddy diffusion coefficient because they do not meet the diffusive criterion, so the eddy scales should be much less than the atmospheric density gradient scale. Turbulence with a large eddy scale sometimes cannot be characterized by $Ri_c$ because of the complicated transition from turbulent to laminar flow. Our results show that $Ri_c$ can undoubtedly be applied to uniform turbulence with small eddies corresponding to the eddy diffusion. It should be emphasized that the direct measurements of the $Ri$ value are impossible and this value is estimated using the experimental data on the temperature and wind velocity. These parameters are very variable during the disturbance and sometimes the observed turbulence does not correspond to the current conditions due to the effect of the previous disturbances, as noted by Bishop et al. [2004].

The vertical size of a few kilometers corresponds to the thickness of the uniform turbulent layers considered by us and corresponds to the experimental data (see, for example, TOMEX). The experimental results presented by Haack et al. [2014] are obtained due to the stratospheric observations of very narrow layers with the typical thickness of 40 m corresponding to the localized turbulence. It is surprising that the reviewer reproached us about the small altitude range but recommends a paper with an altitude range that is smaller by a factor of 50 than the range used by us. In this case, the eddy sizes can be comparable with the thickness of the turbulent layer, and the interaction between the buoyancy force and wind shear can be very complicated. Balsley et al. [2008] showed that $Ri_c$ can only exist when the scales are small. This is in agreement with our conclusion mentioned above. According to Fig. 9 in Haack et al. [2014], the very large buoyancy frequency and the very small wind shear are observed in these turbulent layers. This can indicate the complicated structure of the wind shear (see, for example, Galperin et al. [2007]. Finally, we consider the uniform turbulence corresponding to the criterion for the eddy diffusion [Vlasov and Kelley, 2015] that is very different from the very thin layers of the localized turbulence where scales can be comparable with the thickness of the layer and $Ri_c$ may not have physical meaning. Also note that the conditions in the lower stratosphere considered by Haack et al. [2014] are very different from the mesosphere. First of all, the density in this region is larger by three orders of magnitude than the mesospheric density.

The $Ri_c$ increase from 0.25 to 0.38 increases the turbulent cooling rate by a factor of 1.6 (section 5 in our paper). The coefficient $b = Ri/(P - Ri)$ ($P$ is the Prandtl number equal to 1 for the uniform turbulence) with $Ri = 0.25$ is commonly used in the formula $K_e = b\varepsilon/\omega_B^2$ to estimate the eddy diffusion coefficient where $\varepsilon$ is the energy dissipation rate and $\omega_B$ is the buoyancy frequency. The $Ri_c$ value increases from 0.25 to 0.38, corresponding to the $b$ and $K_e$ value increases by a factor of 2, which means a serious change in the height distributions of the important constituents in the mesosphere and lower thermosphere. The total density in the thermosphere depends on this coefficient because the atomic oxygen height distribution formed by eddy diffusion in the mesosphere becomes the main constituent in the thermosphere.

*List of references contains only 14 items (with 2 from the same authors).*

*That is, a proper overview of previous results regarding Ric is not present. Quick look in 50 years old overview articles [e.g., Reiter and Lester, 1967; Obukhov, 1971] reveals that different values of Ri$_c$ with different approaches were obtained, which is not mentioned in this manuscript. In the manuscript, authors refer to papers of Miles and Howard from 1961 (i.e., 57 years back). Does it mean, that in the last half century nobody considered this problem?*

This reviewer's statement is incorrect. The references in our paper include, for example:

Abarbanel, H.*,* Holm*,* D.*,* Marsden, J., and Ratiu, T.: Richardson number criterion for the nonlinear stability of three-dimensional stratified flow*, Phys. Rev. Lett.*, *52*, 2352–2355*,* 1984.

Galperin, B., Sukoriansky, S., and Anderson, P. S.: On the critical Richardson number in stably stratified turbulence, *Atmos. Sci. Lett.*, 8(3), 65-69, 2007.

Hysell, D. F., Nossa, E, Larsen, M, F., Munro, Smith, J S., Sulzer, M. P. and González, S. A.: Dynamic instability in the lower thermosphere inferred from irregular sporadic E layers, *J. Geophys. Res.*, 117, A08305, doi:10.1029/2012JA017910, 2012.

Ligniéres, F.*,* Califano*,* F., and Mangeney, A.: Shear layer instability in a highly diffusive stably stratified atmosphere, *Astron. and Astrophys.*, 349, 1027-1036*,* 1999.

As can be seen from this list of references, the papers published in 2007 and 2012 are included. Note that the paper by Hysell et al. [2012] is based on the approach developed by Miles [1961]. It is necessary to emphasize that the goal of our paper is the theoretical estimate of the critical Richardson number for the development of dynamic instability in the mesosphere. The large negative gradient of the temperature is the main feature of this region of the upper atmosphere. However, excluding the paper by Hysell et al. [2012], the other papers mentioned above are not related to the mesospheric condition and this is even more so for the papers mentioned by the reviewer. The latter are dealing with the turbulence in the boundary layer where atmospheric gas interaction with the surface of the earth plays a very important role and the conditions are very different from the conditions in the upper atmosphere. The former is the object of *meteorology*,

which studies the lower atmosphere, and the latter is the object of *aeronomy*, which studies the behavior of the free gas in the upper atmosphere.

It is noted in our paper that "we could not find papers on the theoretical estimate of the critical Richardson number that take the mesospheric conditions (first of all, the large negative gradient of the temperature) into account". Unfortunately, the reviewer also could not find papers concerning the problem considered in our paper. Three of the four references given by the reviewer are not concerned with the upper atmosphere because these papers are dealing with the boundary layer in the lower atmosphere. The paper [Haack et al., 2014] mentioned by the reviewer in the reviewer's previous comment was discussed in detail in our response to that reviewer's comment. Note that Haack et al. [2014] quotes the Galperin et al. [2007] paper mentioned in our paper. Finally, we are very surprised that the reviewer does not make a distinction between the atmosphere in the boundary layer and the upper atmosphere.

*Also some reference to the statement "However, the eddy turbulence peak is observed in the mesosphere or the lower thermosphere where the large negative and positive gradients of the temperature occur." (lines 47-49) is needed.*
*Authors concluded, that $Ri_c$ value depends on the temperature gradient. The $Ri_c$ value increases with the negative mesospheric temperature gradient increase. (for example, lines 226-227). These statements were supported by different figures that only show an altitude range above 90 km where temperature gradient is already positive. This is confusing. Is it possible to use altitude range below mesopause for simulations?*

It is well known that the peak of turbulence takes place in the mesosphere and low thermosphere [Brasseur and Solomon, 1986; Fukao, et al., 1994]. First of all, this is a result of the maximum energy dissipation of the gravity waves. The mean energy dissipation rate can be 0.3 – 0.4 W/kg [Bishop et al., 2004] in this region, but this mean rate does not exceed 0.04 W/kg [Haack et al., 2014] in the stratosphere.

The reviewer's confusion is wrong. At first, this confusion was based on a very old schematic diagram of the temperature height distribution in the upper atmosphere (see, for example, Banks and Kockarts [1973]). However, modern schematic diagrams show the mesopause altitude of 95 km (see, for example, Schunk and Nagy [2009]). According to the global experimental data generalized by the empirical models, the mesopause is located at the altitude significantly above 90 km in winter, spring, summer, and autumn seasons at all latitudes and the maximum of the negative gradient of the temperature takes place at around 90 km, excluding the latitude region of 50°N – 70°N in summer where this altitude can be below 90 km. An example of the latitudinal distributions of the mesopause altitude in the summer and autumn equinoxes is shown in Fig. 1, according to the global empirical model MSISE-90 [Hedin, 1991]. Note that the mesopause altitudes shown in Fig. 1 can be considered as the mean values between the winter (99 km) and summer (97 km) values at the middle and low latitudes. Additionally, this can be seen from the

annual mean temperature height profile shown in Fig. 2. Unfortunately, the reviewer is more familiar with the lower atmosphere than the upper atmosphere.

Second, the negative gradient of the temperature in the turbulent layer can be produced due to gravity waves at any altitude in the mesosphere and lower thermosphere. However, the temperature negative gradient in the undisturbed mesosphere provides a more comfortable condition for the production of this gradient in the dynamic instability.

Third, it is necessary to emphasize that the $Ri_c$ dependence on the temperature negative gradient is obtained by us without using the density, neutral composition, and other parameters of the mesosphere (formulas (16) and (17) in our paper). This means that this result can be applied at any altitude where the negative gradient of the temperature takes place in the mesosphere and the lower thermosphere with uniform turbulence.

Unfortunately, the reviewer does not discuss the main points of our paper: a new assumption for the $Ri_c$ estimate, the influence of the temperature negative gradient on the $Ri_c$ value, and the increase in the cooling rate due to this influence.

References

Banks, P. M. and Kockarts, G.: *Aeronomy*, Academic Press, New York, USA, 1973.

Bishop, R. L., Larsen, M. F., Hecht, J. H., Liu, A. Z., and Gardner, C. S.: TOMEX: Mesospheric and lower thermospheric diffusivity and instability layers, *J. Geophys. Res*., 109, D02S03, doi:10.1029/2002JD003079, 2004.

Brasseur, G., and Solomon, S.: *Aeronomy of the Middle Atmosphere*, D. Reidel Publishing Company, 1986.

Fukao, S., et al.: Seasonal variability of vertical eddy diffusivity in the middle atmosphere, 1. Three-year observations by the middle and upper atmosphere radar, *J. Geophys. Res*., vol. 99, 18,973-18,987, 1994.

Gardnera, C. S., Zhao, Y., and Liu, A. Z.: Atmospheric stability and gravity wave dissipation in the mesopause region, *J. Atmos. Solar-Terr. Phys.,* 64, 923–929, 2002.

Hedin, A. E.: Extension of the MSIS thermosphere model into the middle and lower atmosphere, *J. Geophys. Res*., 96, 1159-1172, 1991.

Schunk, R.W., and Nagy, A.F.: *Ionospheres, Physics, Plasma Physics and Chemistry*, second edition, Cambridge University Press, 2009.

Vlasov, M. N., and Kelley, M. C.: Eddy diffusion coefficients and their upper limits based on application of the similarity theory, *Ann. Geophys*., 33, 857-864, 2015.

Weinstock, J.: Vertical turbulent diffusion in a stably stratified fluid, *J. Atmos. Sci.*, 35, 1022–1027, 1978.

[Figure]

Fig. 1. The latitudinal variations of the mesopause altitude given by the MSISE-90 model [Hedin, 1991] for the autumn and spring equinox (dashed and solid curves, respectively).

[Figure]

Fig. 2. Annual mean temperature height profile derived from more than 1000 h of Na temperature lidar observations obtained throughout the year and the diurnal cycle at the Urbana Atmospheric Observatory (40°N, 88°W) [Gardner et al., 2002].

*Reviewer's References*

*Grachev, A., E. L Andreas, C. Fairall, P. Guest, and O. Persson, The critical richardson number and limits of applicability of local similarity theory in the stable boundary layer, 147:51–82 147, 2012. Boundary-Layer Meteorology, v. 147, p. 52-82, 2013*

*Haack, A., M. Gerding, and F.-J. L• ubken, Characteristics of stratospheric turbulent layers measured by LITOS and their relation to the Richardson number, J. Geophys. Res., pp. 10,605{10,618, doi:10.1002/2013JD021008, 2014.*

*Obukhov, A. M., Turbulence in an atmosphere with a non-uniform temperature, Boundary-Layer Meteorology, 2, 7{29, doi:10.1007/BF00718085, 1971.*

*Reiter, E. R., and P. F. Lester, The Dependence of Richardson Number on Scale Length, Colorado State University, Atmospheric Science, 111, 39, 1967.*

---

## Referee Comment (RC2) · Anonymous Referee #2 · 20 Nov 2018

In this manuscript the authors study the dependence of the critical Richardson number on the temperature gradient by rewriting the buoyancy frequency and wind shear terms to be dependent solely on the temperature. They evaluate the critical Richardson number for isothermal atmosphere and for temperature decreasing with altitude. At this stage, the level of the study is poor. It has to be completely rewritten and undergo a new round of reviews to be assessed for publication. Regarding the title, I completely do not get the relationship to the mesosphere, besides the fact that the authors also consider situations with negative vertical gradient of temperature. Ref#1 also raised this issue and in the AC comment the authors contradict themselves by arguing (on three full pages) that the other studies (like Obukhov (1971)) are not appli-

cable for the mesosphere, but then surprisingly in the final paragraph they write:" ...Ric dependence.....is obtained by us without using density, neutral composition, and other parameters of the mesosphere.." With some weird remark that the applicability is linked with the uniform turbulence. Btw. the study of Obukhov (1971) gives a rigorous summary of the Ri and Ric dependence on the temperature gradient and the author need to explicitly cite this study and show, where they give superior scientific information. The connection of the study under review with the mesosphere is demonstrated by the figures, where the x axis shows height about 90 km. But, this is just due to the author arbitrariness connected probably with the choice of temperature values they used for evaluations.

MAJOR CONCERN: A)Most importantly, I have serious concern about the validity of the methodology and flawlessness of the analytical derivations in this paper: The crucial point of this study is that the authors assume adiabatic expansion. While this can be a good assumption for the GW induced perturbations, it is completely irrelevant for the background, where e.q. the solar tidesgovern a significant part of the mesospheric variability. Also, the authors use this assumption to connect the vertical gradient of full (background + disturbed) density distribution to the full temperature and its gradient and wind shear (Eqs. 6,7,8,9, 10). Also in the light of tides, this assumption crucial for the paper needs to be properly justified, ideally by referencing observational studies. But more than just general doubts about the validity of this assumption, the authors make errors also in analytical description, where in eq. 8, which shows partial derivative of T with altitude they refer to it as (P4L81) "temperature gradient in the parcel (sic) with upward motion and adiabatic expansion" - but for this, total derivative would have to be shown. Most importantly, on their way from eg. 6 to 10 they use in P4L80 an equation for Ri based on different assumption (they don't tell anything about this formula, which is crucial) and then they consider this Ri (general?) to be equal to the Ri in eq. 7 (adiabatic expansion) for deriving eq. (10). A similar situation takes place in section 3, where they give equation 13b (P6L110) without properly discussing how they derived this equation and the underlying assumptions (polytropic atmosphere?). This

formula (13b) and the formula for wind shear (eq. 10) are the crucial parts of the paper, because every other result then presented is only a trivial evaluation of Ri based on those formulas. The authors need to carefully rewrite all of their analytical derivations, distinguish properly between local and total derivatives, list the assumptions made and ensure consistency between the assumptions and also distinguish in their formulas between constants and functions of altitude (f(z)). Without this it doesn't make sense to discuss any results given later in the text (poor evaluation of the derived formulas), because my personal opinion (the authors are welcomed to prove otherwise) is that the results are dominated by flaws in their analytical construct.

B) Language: Non-scientific language is used frequently, with weird phrases like: we could find just one paper..or the authors write that some study is wrong, but do not prove it. Just to list: What is the acceleration in wind shear? P5L92 Does wind shear really induce vertical accelerations? (no, you have to replace the word induce by e.g. support) Page 3, L 67 not wind shear nor stability are forces.. Those were the most striking ones. I am not listing all the typos made in the manuscript because I expect major changes before it can be assessed for publication.

---

## Author Comment (AC2) · 18 Dec 2018

**Author Response to the Comments of Reviewer 2:**

"Dependence of the critical Richardson number on the temperature gradient in the mesosphere"

Michael N. Vlasov and Michael C. Kelley

**Reviewer comment:**

Regarding the title, I completely do not get the relationship to the mesosphere, besides the fact that the authors also consider situations with negative vertical gradient of temperature.

**Author response:**

The object of our study is turbulence in the mesosphere as the region of the upper atmosphere that includes the stratosphere, mesosphere, and thermosphere. The most import feature of the stratosphere and thermosphere is the positive gradient of the temperature. The mesosphere is the only region in the upper atmosphere that is characterized by the temperature negative gradient. The other main features of the mesosphere are the turbulence peak in the upper mesosphere and the wind shear maximum in this region. The negative gradient of the background temperature and the wind shear (Larsen, 2002) (page 5, lines 95-97) in this region provide sufficient and essential conditions for the development of dynamic instability. Turbulence and wind shear are not observed in the thermosphere. According to the observations (for example, Haack et al. (2014)), the very narrow layers of turbulence (localized turbulence) take place in the stratosphere. According to Fig. 9 in Haack et al. (2014), a very large buoyancy frequency and a very small wind shear are observed in these turbulent layers. This indicates the complicated structure of the wind shear (see, for example, Galperin et al. (2007)). Our assumption cannot be applied in this case because of the problem with determining the acceleration of this wind shear.

Revision in the paper: no changes

**Reviewer comment:**

Ref#1 also raised this issue and in the AC comment the authors contradict themselves by arguing (on three full pages) that the other studies (like Obukhov (1971)) are not applicable for the mesosphere, but then surprisingly in the final paragraph they write:" ...Ric dependence...is obtained by us without using density, neutral composition, and other parameters of the mesosphere." With some weird remark that the applicability is linked with the uniform turbulence. The connection of the study under review with the mesosphere is demonstrated by the figures, where the x axis shows height about 90 km. But, this is just due to the author arbitrariness connected probably with the choice of temperature values they used for evaluations. Author response:

Obukhov considers the turbulence in the surface layer. He notes that, "Since the height of the surface layer is not great (on the order of a few tens of meters), the changes of absolute density and temperature within the layer are small and can be considered negligible". This means neglecting the term  $(p_0/p)^{\xi}$  with altitude in the formula  $\theta = T(p_0/p)^{\xi}$  and using the formulas

$$\frac{\partial \theta}{\partial z} = \left(\frac{\partial T}{\partial z} + \frac{g}{C_p}\right) \qquad Ri = \frac{g}{T} \frac{\partial \theta}{\partial z} \left(\frac{\partial v}{\partial z}\right)^{-2}$$

This approach and these formulas cannot be used for the mesosphere. The thickness of the surface layer considered in the paper is less by a factor of 80 than the scale height of the atmosphere (about 8 km) and this condition is very different from the mesospheric conditions where the scale heights of 4 - 6 km and the thickness of the turbulent layers may be larger than 1 km and the turbulence occupies a region of 40 km. Also, there are other important distinctions between the surface layer in the lower troposphere and the mesosphere. Apparently, the reviewer does not know the principal distinctions between the surface layers in the lower troposphere.

Revision in the paper: no changes

**Reviewer comment:**

Btw. the study of Obukhov (1971) gives a rigorous summary of the Ri and Ric dependence on the temperature gradient and the authors need to explicitly cite this study and show where they give superior scientific information.

**Author response:**

This reviewer's statement is wrong. There is only one sentence on estimating the Ricr value in the paper (page 15): "Corresponding processing of Sverdrup's data leads to  $Ri_{cr} = 1/11$ , which is used later in numerical calculations" and then the author states that, "The determination of the critical *Ri* number is an important problem for atmospheric physics and may be solved only experimentally on the basis of processing reliable data for simultaneous measurements of wind and temperature distributions in the lower layer of the atmosphere".

Thus, Obukhov uses the experimentally determined value (the only value) of  $Ri_c$  for a very rough estimate of the temperature gradient according to his statement (page 21): "Thus, the order of magnitude of the temperature gradient calculated according to  $K_{\infty}$  agrees with the observations. In accordance with Sverdrup's observations, the value  $Ri_c = 1/11$  was used during calculations of the gradient". It is necessary to emphasize that no dependence of the  $Ri_c$  value on the temperature gradient is presented because the author used the only value of  $Ri_c = 1/11$  that was experimentally determined. This is exactly the opposite of what we have done in our paper. We theoretically define

the  $Ri_c$  value and calculate the different  $Ri_c$  values for the different temperature gradients (see Figs. 3b and 4).

It is necessary to emphasize that Obukhov's result with a huge uncertainty in the temperature gradient calculated for the Ric fixed value strongly contradicts the direct and unique dependence of the Ric value on the temperature gradient presented in our paper. This contradiction and other problems with estimates using some formulas presented in the paper are explained in the paper by A.S. Monin and A.M. Obukhov, "Turbulent mixing in the atmospheric surface layer" (*Trudy Geophys. Inst.*, 1954, N°24, 151 and "*Turbulence and atmospheric dynamics*", ed. J.L. Lumley, NASA, CTR Monograph, November 2001, p. 164). The authors of this paper state that "Obukhov used some insufficiently reliable data (the critical Richardson number was erroneously taken to be 1/11 on the basis of Sverdrup's results) and therefore we could not directly apply his formulas for the practical calculations". This statement is in good agreement with our attempt to use some of the formulas given in Obukhov's paper.

We are very confused by the reviewer's recommendation of this paper, which, according to the author's statement in his next paper, presents the wrong  $Ri_c$  value and the wrong formulas are used.

It should be noted that Obukhov's paper was published in 1946 by the journal *Trudy Inst Teor. Geophys*, vol. 1, 95–115. However, this publication was really inaccessible outside of the USSR. The reference given by reviewer 2 corresponds to a translation of this paper published by the journal *Boundary-Layer Meteorol*, 1971, 2, 7-29. In the introduction to this publication, J. A. Businger and A.M. Yaglom explain the reason for this publication: "Probably the major contribution of the paper is the introduction of the 'length scale of the dynamic turbulence sublayer', *L*. This length scale was later introduced independently by Lettau (1949), and at present, it is commonly known as the Monin-Obukhov length. Its fundamental role in the whole field of boundary-layer meteorology was most clearly explained in the well-known paper by Monin and Obukhov (1954)". The authors of the introduction do not mention the problem with the Richardson number in Obukhov's paper because of the comments in Monin and Obukhov (1954) discussed above.

Revision in the paper: no changes

**Reviewer comment:**

A) Most importantly, I have serious concern about the validity of the methodology and flawlessness of the analytical derivations in this paper: The crucial point of this study is that the authors assume adiabatic expansion. While this can be a good assumption for the GW induced perturbations, it is completely irrelevant for the background, where e.g., the solar tides govern a significant part of the mesospheric variability. Also, the authors use this assumption to connect the vertical gradient

of full (background + disturbed) density distribution to the full temperature and its gradient and wind shear (Eqs. 6,7,8,9, 10). Also in the light of tides, this assumption crucial for the paper needs to be properly justified, ideally by referencing observational studies.

Author response:

"Turbulence is generated by waves breaking in the MLT through mechanisms such as convective and dynamic instabilities (e.g., Hodges, 1969; Lindzen, 1981; Zhao et al., 2003; Liu et al., 2004; Williams et al., 2006; Hecht et al., 2014)". Hecht, J. H., K. Wan, L. J. Gelinas, D. C. Fritts, R. L. Walterscheid, R. J. Rudy, A. Z. Liu, S. J. Franke, F. A. Vargas, P. D. Pautet, M. J. Taylor, and G. R. Swenson ((2014); The life cycle of instability features measured from the Andes Lidar Observatory over Cerro Pachón on 24 March 2012, *J. Geophys. Res. Atmos.*, 119, 8872–8898, doi:10.1002/2014JD021726". (Guo, Y., A. Z. Liu, and C. S. Gardner (2017), *Geophys. Res. Lett.*, 44, 5782–5790,doi:10.1002/2017GL073807.)

Hodges (*J. Geophys. Res.*, 72, 3455-3458, 1967) pointed out that it is unlikely to have conditions for dynamic instability without gravity waves. Tides alone are not sufficient to induce dynamic or convective instabilities, but the tides can influence the conditions for dissipation of the gravity waves and the development of dynamic instability due to change in the temperature gradient. In any case, adiabatic expansion is a fundamental process for dynamic instability and the adiabatic lapse rate is a very important parameter. This assumption is used to derive the buoyancy frequency formula (see, for example, Peixoto, J. P., and Oort, A. H.: *Physics of Climate*. New York: Springer-Verlag, 1992), which is included in the chain of equations (6)–(10). The Richardson number depends directly on the adiabatic lapse. Unfortunately, the reviewer does not explain why adiabatic expansion cannot exist for the tides. We do not consider the mesospheric background parameters' variability induced by the different processes. We only consider the dependence of dynamic instability on the temperature gradients in the mesosphere. Unfortunately, the reviewer does not explain what kind of observational studies he means. In our paper, the results of the experimental data (Bishop et al., 2004; Kelley et al., 2003; Larsen, 2002; Lubken, 1997) are used.

Revision in the paper: no changes

**Reviewer comment:**

But more than just general doubts about the validity of this assumption, the authors make errors also in analytical description, where in eq. 8, which shows partial derivative of T with altitude they refer to it as (P4L81) "temperature gradient in the parcel (sic) with upward motion and adiabatic expansion" - but for this, total derivative would have to be shown.

**Author response:**

We are very surprised by this comment. Eq. 8 is the result of the simple combination of generally accepted Eqs. 2, 6, and 7 with partial derivatives and it is impossible to obtain this formula with total derivatives in only one equation in this combination. Eq. 6 is the key formula and presents the temperature gradient corresponding to adiabatic expansion due to upward parcel displacement. This result does not depend on the kinetics of parcel motion. This is the generally accepted approach for estimating the effect of parcel displacement on the temperature for adiabatic expansion/compression. Unfortunately, the reviewer's statement is too general without an explanation or a reference.

Revision in the paper: no changes

**Reviewer comment:**

Most importantly, on their way from eq. 6 to 10 they use in P4L80 an equation for Ri based on different assumption (they don't tell anything about this formula, which is crucial) and then they consider this Ri (general?) to be equal to the Ri in eq. 7 (adiabatic expansion) for deriving eq. (10).

**Author response:**

The derivation of Eq. 6 was given in Appendix 1. Taking this comment into account, an additional explanation is included in the text (page 3) and Appendix 1. The main point is that Eq. 4 corresponds to incompressible fluid and  $\omega_B^2 = (-g/\rho_0)\partial\rho_0/\partial z$ , but Eq. 6 corresponds to compressible fluid (adiabatic expansion) and  $\omega_B^2 = (g/T)(\partial T/\partial z + g/Cp)$  should be used, so in this case, Eq. 7 and Eq. 8 must correspond to compressible fluid.

Revision in the paper: page 3, lines – 57, 59, 63 page 4, lines 72, 73, 80, 81, 83, 90 page 16, lines – 262, 263, 264, 266

**Reviewer comment:**

A similar situation takes place in section 3, where they give equation 13b (P6L110) without properly discussing how they derived this equation and the underlying assumptions (polytropic atmosphere?). This formula (13b) and the formula for wind shear (eq. 10) are the crucial parts of the paper, because every other result then presented is only a trivial evaluation of Ri based on those formulas.

Author response:

We did not show the derivation of formula (13b) because this formula is the same as the well-known and commonly used formula (Banks and Kockarts, 1973, part A, page 36, 1973):

 $\rho = \rho_0 \left( H / H_o \right)^{-(1+\beta)/\beta}$ (A1) where  $H = \kappa T/mg$ ,  $\alpha = \beta = \partial H/\partial z = (\kappa/mg)\partial T/\partial z$ , and  $n = \rho/m$ .

The derivation of eq. 13b is now given in Appendix 4.

Revision in the paper: Page 19, lines 322-340.

**Reviewer comment:**

The authors need to carefully rewrite all of their analytical derivations, distinguish properly between local and total derivatives, list the assumptions made an ensure consistency between the assumptions and also distinguish in their formulas between constants and functions of altitude (f(z)). Without this it doesn't make sense to discuss any results given later in the text (poor evaluation of the derived formulas), because my personal opinion (the authors are welcomed to prove otherwise) is that the results are dominated by flaws in their analytical construct.

**Author comment:**

The reviewer's negative comments are too general without any evidence, examples, or references. For instance, the reviewer says that "the results are dominated by flaws" but does not prove his/her mere allegations. Moreover, the reviewer has stated (in two separate instances) that the assumptions have not been explicitly listed in the paper, whereas in fact, they were provided on pages P2L27,28; P3L58-64; P4L72,73; P5L96,97; P6L 111-113; P7L114,115 and L126,127; and P13L204-206 of the submitted manuscript. Also, it is totally unclear why the reviewer insists on using "total derivatives" while all the well-known formulas are customarily defined in terms of partial derivatives.

Revision in the paper: no changes

**Reviewer comment:**

Language: Non-scientific language is used frequently, with weird phrases like: we could find just one paper... or acceleration in wind shear the authors write that some study is wrong, but do not prove it. Just to list: What is the? P5L92 Does wind shear really induce vertical accelerations? (no, you have to replace the word induce by e.g., support) Page 3, L 67 not wind shear nor stability

are forces. Those were the most striking ones. I am not listing all the typos made in the manuscript because I expect major changes before it can be assessed for publication.

Author response:

Note that reviewer 1 did not have a problem with the language used in our paper. We made a few language corrections to the text. The reviewer's statement, "the authors write that some study is wrong, but do not prove it," is incorrect. The explanation was presented in detail in Appendix 3. Note that this reviewer's statement does not demonstrate a language problem. Our paper stated, "The goal of this paper is to estimate the critical Richardson number,  $Ri_c$ , corresponding to the equilibrium between the buoyancy force and the force induced by wind shear in the mesosphere. Dynamic instability is developed for  $Ri < Ri_c$ . Our approach considers the acceleration corresponding to both forces, taking into account the mesospheric temperature height distributions". It is not clear why the reviewer objects to the word "force". Again, note that reviewer 1 did not have a problem with the language used in our paper.

In general, reviewer 2's apparent lack of understanding concerning the distinction between the surface layer in the troposphere and the mesosphere, the unproven statements about the important role of tides for dynamic instability development, the use of total derivatives in commonly used formulas, and his/her request to present the derivation of the well-known and commonly used formula of density distribution in the upper atmosphere clearly demonstrate that the reviewer is not adequately familiar with the physics of the upper atmosphere and dynamic instability. One obvious evidence of this is the reviewer's persistent recommendation of a paper that, according to the author's statement in his next paper, presents the wrong  $Ri_c$  value and uses the wrong formulas.

Revision in the paper: Changes were made, including on page 3.

**Dependence of the critical Richardson number on the temperature gradient in**

**2 the mesosphere**

3 Michael N. Vlasov and Michael C. Kelley

[revised manuscript text omitted]